

# Genetic diversity and population structure of muscovy duck (*Cairina moschata*) from Nigeria

Adeniyi C. Adeola[1,2,3], Foluke E. Sola-Ojo[4], Yusuf A. Opeyemi[4], Abel O. Oguntunji[5], Lotanna Micah Nneji[6], Muslim K. Ewuola[7], Semiu F. Bello[8], Wasiu A. Olaniyi[9], Adeosun T. Adesoji[10], Alex P. Karuno[1], Oscar J. Sanke[11] and Ebiakpo Lucky Daniel[12]

[1] Molecular Evolution and Genome Diversity, State Key Laboratory of Genetic Resources & Evolution, Kunming Institute of Zoology, Chinese Academy of Sciences, Kunming, Yunnan, China
[2] Sino-Africa Joint Research Center, Chinese Academy of Sciences, Sino-Africa Joint Research Center, Chinese Academy of Sciences, Kunming, Yunnan, China
[3] Centre for Biotechnology Research, Bayero University, Kano, Nigeria
[4] Department of Animal Production, Faculty of Agriculture, University of Ilorin, Ilorin, Kwara, Nigeria
[5] Department of Animal Science and Fisheries Management, Bowen University, Iwo, Osun, Nigeria
[6] Department of Ecology and Evolutionary Biology, Princeton University, Princeton, New Jersey, United States
[7] Animal Breeding and Genetics Unit/Department of Animal Science, University of Ibadan, Ibadan, Oyo, Nigeria
[8] Department of Animal Genetics/Breeding and Reproduction/College of Animal Science, South China Agricultural University, Guangzhou, China
[9] Department of Animal Science, Faculty of Agriculture, Adekunle Ajasin University, Akungba Akoko, Ondo State, Nigeria
[10] Department of Agricultural Education, Federal College of Education, Bichi, Kano, Nigeria
[11] Taraba State Ministry of Agriculture and Natural Resources, Jalingo, Taraba, Nigeria
[12] Department of Animal Science, University of Benin, Benin City, Nigeria

Corresponding author
Adeniyi C. Adeola,
chadeola@mail.kiz.ac.cn

## ABSTRACT

The domestic Muscovy duck (*Cairina moschata*) provide unique genetic resources patterned by both tropical environmental conditions and human activities, the evaluation of their genetic diversity and population structure will shade light on the mechanism of their remarkable adaptive capacities. We therefore analyzed the variation in mtDNA cytochrome b and nuclear DNA CYP2U1 sequences of 378 Nigerian Muscovy ducks (comprising of 287 *de novo* and 91 downloaded) plus 80 published sequences of Muscovy ducks from India. The results showed high haplotype diversity (0.800 ± 0.023) among Nigerian Muscovy duck populations with 91 distinct haplotypes for the nuclear DNA CYP2U1 gene but low (0.266 ± 0.033) for cytochrome b with 31 haplotypes. The median-joining networks of both markers grouped Nigerian Muscovy ducks into two; the first group consisting of only Nigerian Muscovy duck populations, and the second group Nigerian with Indian populations. Neutrality test results indicated that Nigerian populations experienced recent population expansion and/or genetic hitchhiking. A geographic signal was absent in line with previously studied poultry species in Nigeria. The most prominent haplotype dominated across all regions in Nigeria, which may be due to extensive genetic intermixing except for the Indian population ($F_{ST}$ = 0.02550, $P$ = 0.01075).
This indicated low genetic differentiation between and within Nigerian Muscovy duck as revealed by the suitability of the nuclear DNA CYP2U1 gene.

# INTRODUCTION

Muscovy duck (*Cairina moschata*) is indigenous to the tropical regions of Central and South America (*Rodenburg et al., 2005*; *Adeola et al., 2020*), and was introduced to other continents of Asia, Africa, and Europe (*Arias-Sosa, Alex & Rojas, 2021*). An inventory survey of livestock in Nigeria revealed that duck is the third most widely domesticated poultry birds (*Hassan & Mohammad, 2003*; *Nwanta et al., 2006*) with an estimated population of 9,553,911 after chicken (101,676,710) and guinea fowl (16,976,907) (*NBS, 2012*). It is noteworthy that reports of studies conducted in northern (*Duru et al., 2006*), western (*Oguntunji & Ayorinde, 2015a*) and eastern (*Ahaotu et al., 2017*) Nigeria unequivocally indicated that Muscovy duck popularly known as local duck was the prevalent genus in Nigeria and predominantly managed extensively. This accounted for about 10% of the local poultry population and 74% of ducks reared in Nigeria (*Adesope & Nodu, 2002*; *Yakubu, 2013*).

Domestic Muscovy ducks are valued throughout the world for their unique flavourful taste, high yield of breast meat, low-calorie content meat, rapid growth rate, good foraging, incubation behaviour, as well as for being less exigent for feed quality and less susceptible to diseases than chicken (*Duru et al., 2006*; *Chen, He & Liu, 2009*; *Raji, Igwebuike & Usman, 2009*). However, with these innate potentials of ducks in Nigeria, genetic studies on their characterization and genetic improvement are insufficient (*Oguntunji & Ayorinde, 2014*; *Oguntunji & Ayorinde, 2015b*). The study of genetic diversity within and between populations is a prerequisite for sustainable utilization of domestic species. Recent advances in molecular genetics have generated detailed genetic information on animals with high accuracy compared to data obtained by pedigree relationship and trait phenotypes. Hence, comprehensive information on the genetic diversity and structure of Muscovy duck is pivotal to its characterization, management, and further improvement.

Studies have investigated diversity in Nigerian Muscovy ducks at morphological, biochemical, physiological, and phenotypic levels (*Okeudo, Okoli & Igwe, 2003*; *Oguntunji & Ayorinde, 2015b*, *2015c*; *Ewuola et al., 2020*). Though these methods are low cost and parameters easier to measure, with less accuracy and precision for selective breeding and commercial selection. Application of molecular genetics has enhanced in-depth characterization and identification of molecular marker-related genes such as single nucleotide polymorphisms (SNPs), that may be applicable in marker-assisted selection (*Baena et al., 2018*).

Several molecular markers have been developed to study the genetic diversity within and between populations, including cytochrome b gene (*Sun et al., 2012*), cytochrome

P450 CYP2 family (*Kubota et al., 2011*), cytochrome P450 2U1 (CYP2U1) (*Kameshpandian, Thomas & Nagarajan, 2018*), mitochondrial D-loop (*Gaur et al., 2018*; *Adeola et al., 2020*; *De et al., 2021*). Cytochrome b gene has been reported as an efficient tool due to its high power of discrimination for species identification and characterization in taxonomy and forensic science (*Kuwayama & Ozawa, 2000*; *Saif et al., 2012*), and has also been used to study molecular evolution (*Prusak & Grzybowski, 2004*). The nuclear DNA CYP2U1 gene is highly conserved and has been confirmed as a suitable marker in the study of poultry genetic diversity (*Kameshpandian, Thomas & Nagarajan, 2018*). Interestingly, cytochrome b and the nuclear DNA CYP2U1 are highly conserved in avian species (*Nelson et al., 2004*; *Devos et al., 2010*; *Sun et al., 2012*). In addition, variation in mitochondrial DNA (mtDNA) D-loop has been commonly used in the evaluation of diversity and genogeographic structure of domestic animals (*Wang et al., 2014*). These markers have revealed inter- and intra-species variation in avian species (*He et al., 2008a*; *Thomson, Gilbert & Brooke, 2014*; *Awad, Khalil & Abd-Elhakim, 2015*; *Kameshpandian, Thomas & Nagarajan, 2018*; *Sola-Ojo et al., 2021*). Previous studies on the genetic analysis of Muscovy duck including *Sun et al. (2012)* revealed limited genetic diversity in Chinese domestic Muscovy duck population. Further, analysis of mtDNA cytochrome b and nuclear DNA CYP2U1 genes showed a low genetic diversity among Indian populations (*Kameshpandian, Thomas & Nagarajan, 2018*). Similarly, *Ogah et al. (2017)* and *Adeola et al. (2020)* reported low genetic diversity in Nigerian Muscovy duck populations based on mitochondrial D-Loop. A recent study using the nuclear DNA CYP2U1 gene suggested extensive genetic intermixing in the studied Nigerian Muscovy duck populations, but limitation due to the sample size hindered further conclusions (*Sola-Ojo et al., 2021*). In addition, the polymorphism in Nigerian Muscovy duck cytochrome b gene has not been reported. Therefore, the objective of this study was to evaluate the variation in mtDNA cytochrome b and nuclear DNA CYP2U1 genes of different Muscovy duck sampling sites, new and published data, in Nigeria. Additionally, we included data of individuals from India, USA, and China retrieved from public database which represent Muscovy duck populations living outside West Africa to understand the genetic relationship of Nigerian Muscovy duck with other Muscovy ducks. The results from this study gave more information on biodiversity and will assist in designing a suitable breeding plan for the improvement of Muscovy duck population in Nigeria.

# MATERIALS AND METHODS

## Ethical considerations

All experimental procedures in the present study were performed in accordance with the Research Guidelines for the Institutional Review Board of Kunming Institute of Zoology, Chinese Academy of Sciences (SMKY-20160105-11) and the University Ethical Review Committee, University of Ilorin, Nigeria (UERC/ASN/2021/2161).

## Sampling and data collection

We sampled a total of 287 domestic Muscovy ducks (155 males and 132 females) from farmer's flocks in six different States in Nigeria as follows (Fig. 1A; Table S1); Sokoto State

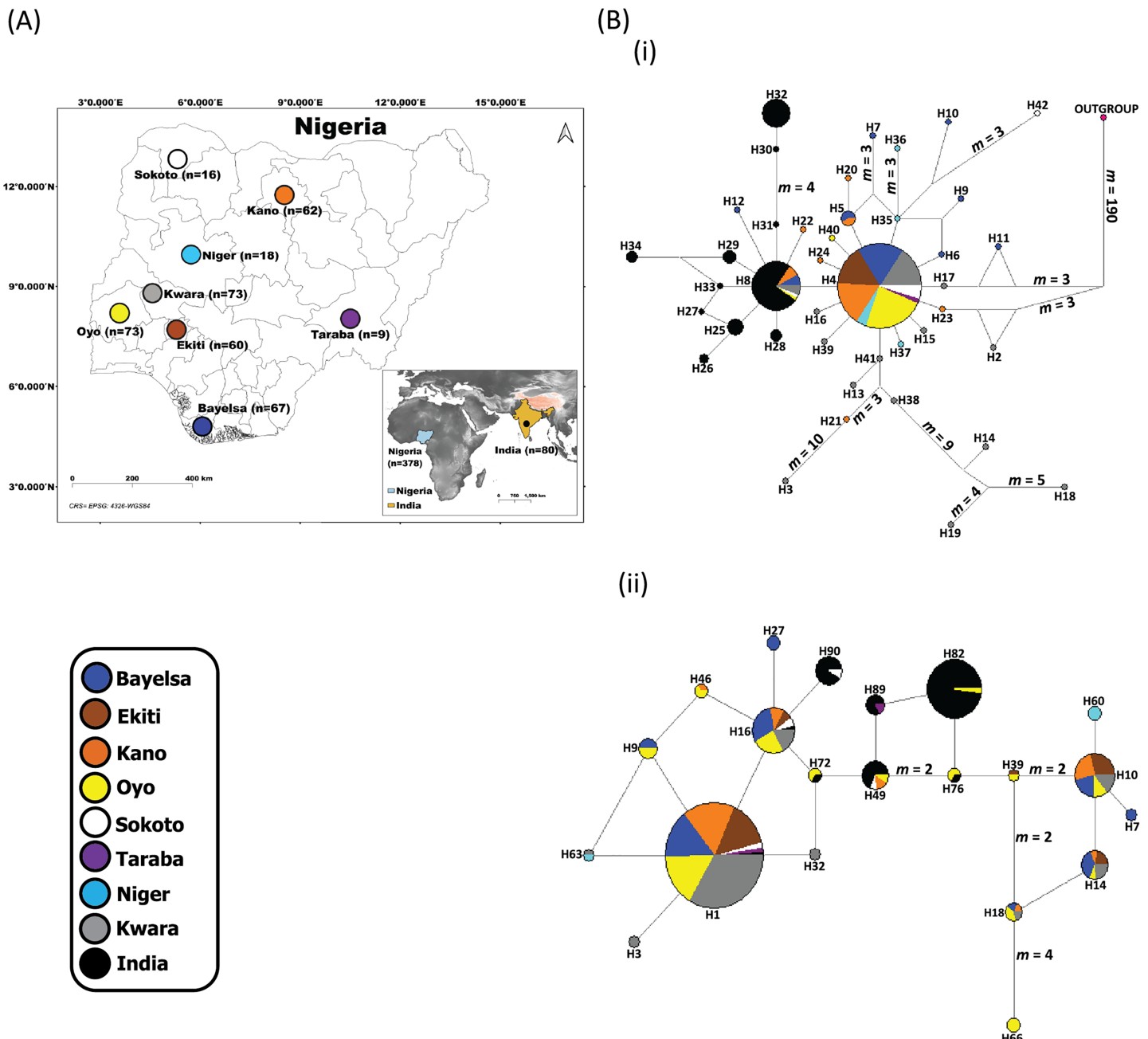

**Figure 1 Sampling locations of domesticated Muscovy ducks and the networks of (i) 397 cytochrome b and (ii) 396 CYP2U1 sequences of domesticated Muscovy ducks based on 940 bp of the cytochrome b and 747 bp of CYP2U1 genes respectively.** (A) Map of 378 domesticated Muscovy duck sampling locations in Nigeria (Sokoto, *n* = 16 from North West; Kano, *n* = 62 from North West; Taraba, *n* = 9 from North East; Oyo, *n* = 73 from the West; Ekiti, *n* = 60 from South West; Bayelsa, *n* = 67 from South South; Kwara, *n* = 73 from North Central; Niger, *n* = 18 from North Central) and India, *n* = 80. Maps of sampling areas shown in color filled circles with the names denoting states and country. The maps were generated using QGIS 3.16.7; (*QGIS Development Team, 2021*). The shapefiles were downloaded from free online DIVA GIS country shapefiles (https://www.diva-gis.org/gdata). (B) Median-joining networks of (i) 397 cytochrome b and (ii) 396 CYP2U1 sequences of domesticated Muscovy duck samples constructed using NETWORK version 4.6 (*Bandelt, Forster & Röhl, 1999*). Reference sequences used for haplotype network analysis included: (i) 397 cytochrome b sequences consisting of Nigeria, *n* = 248 (*de novo*); Niger, *n* = 14 (*Sola-Ojo et al., 2021*); Kwara, *n* = 57 (*Sola-Ojo et al., 2021*) and India, *n* = 78 (*Kameshpandian, Thomas & Nagarajan, 2018*). (ii) 396 CYP2U1 consisting of Nigeria, *n* = 237

**Figure 1 (continued)**
(*de novo*); Niger, *n* = 15 (*Sola-Ojo et al., 2021*); Kwara, *n* = 64 (*Sola-Ojo et al., 2021*) and India, *n* = 80 (*Kameshpandian, Thomas & Nagarajan, 2018*). Sizes of the circles are proportional to haplotype frequencies. *m*, refers to number of mutation steps and those not indicated are just one step mutation. Colours indicate the geographical distribution of the sampling locations across Nigeria and India as shown by the legend in (A).

($n$ = 7 males; $n$ = 9 females), Kano State ($n$ = 36 males; $n$ = 26 females), Taraba State ($n$ = 4 males; $n$ = 5 females), Oyo State ($n$ = 44 males; $n$ = 29 females), Ekiti State ($n$ = 26 males; $n$ = 34 females), and Bayelsa State ($n$ = 38 males; $n$ = 29 females). Our sample collection considered genetically unrelated domestic Muscovy ducks with geographical and ecological perspectives from the six states in Nigeria. This included random sampling of at least two matured birds per flock from domestic Muscovy ducks raised on free-range scavenging system with a distance of at least 500 m apart. Information on the pedigree of the birds were obtained from their owners before blood collection. Blood samples were collected from each bird through their wing venipuncture with the assistance of trained veterinary personnel. Blood samples were preserved in 95% ethanol at room temperature thereafter transported to the laboratory and stored at 4 °C for instant use, or −80 °C for extended storage.

## DNA extraction, PCR and sequencing

Genomic DNA was extracted from whole blood using the standard phenol/chloroform method (*Sambrook & Russell, 2001*). We examined genomic DNA for both quality and quantity using 2% Agarose Gel Electrophoresis against a 2 kb DNA ladder marker and Thermo Scientific™ NanoDrop 2000 spectrophotometer respectively.

Two overlapping primers were used to amplify 940 bp of cytochrome b: forward for the first set DUKCYB1L (5′-ATCTTTCGCCCTATCCATCC-3′) and reverse DUKCYB2R (5′-TTTGGTTTACAAGACCAATGTTTT-3′), and the second, P105 (5′-GCCTCCTGCTAGCCATACAC-3′) and P106 (5′-TACGGCGGGAAAGAGAAATA-3′) (*Kameshpandian, Thomas & Nagarajan, 2018*). The cytochrome b amplification reaction was performed in a 50 μl PCR mixture containing 5 μl 10× reaction buffer, 1.5 mM MgCl$_2$, 0.2mM dNTPs, 0.2 μM each primer, 1.5 U Takara Taq DNA polymerase, and approximately 30 ng genomic DNA. The amplification conditions: 95 °C for 5 min, 30 cycles at 94 °C for 30 s, 54 °C for 45 s, 1 min for 72 °C and a final extension of 72 °C for 5 min (*Kameshpandian, Thomas & Nagarajan, 2018*). After, the quality of PCR products was confirmed on 2% agarose gel.

In addition, 747 bp fragment of nuclear DNA CYP2U1 was amplified using forward primer P103 (5′-GTTATTTGGTTATGCATATCGTG-3′) and reverse primer P104 (5′-GAGACGGTTGGCGTATATGG-3′) (*Kameshpandian, Thomas & Nagarajan, 2018*). Similar PCR conditions as above was used except for annealing temperature of 45 °C. The same quality check on 2% agarose gel was done for nuclear DNA CYP2U1 PCR products. The amplified cytochrome b and CYP2U1 fragments were purified with Exo-SAP-IT cleanup kit as per manufacturer's instructions (Affymetrix). Sequencing reactions were performed in both directions using the BigDye™ Terminator Cycle Sequence Kit 3.1

Ready Reaction Cycle Sequencing Kit (ABI Applied Biosystems, Waltham, MA, USA) and the products afterwards were purified through alcohol precipitation. Finally, ABI PRISM 3,730 automated DNA sequencer (ABI Applied Biosystems, Waltham, MA, USA) was used to analyze the purified products. We assembled all sequences in SEQMAN PRO of LASERGENE 7.1.0 (DNAStar Inc., Madison, WI, USA).

## Data analysis

### Sequences check and alignment

The 248 cytochrome b and 237 nuclear DNA CYP2U1 sequences were mapped to the reference sequences of cytochrome b (GenBank accession no. L08385; *Kornegay et al., 1993*) and *Anas platyrhynchos* cytochrome P (450 2U1XM_013099877) using SeqMan Pro of Lasergene 7.1.0 (DNAStar Inc., Madison, WI, USA). Further, 149 cytochrome b sequences of Muscovy ducks (India (*n* = 78) KX985658–KX985735; Nigeria (*n* = 71) MZ330001–MZ330071) and 159 nuclear DNA CYP2U1 sequences (India (*n* = 80) KX985578–KX985657; Nigeria (*n* = 79) MZ383446–MZ383524) from India and Nigeria in previous studies (*Kameshpandian, Thomas & Nagarajan, 2018*; *Sola-Ojo et al., 2021*) were mined from public database for further study. Multiple alignments of 397 cytochrome b (248 *de novo* [GenBank: MZ383528–MZ383776] and 149 from public database) and 396 nuclear DNA CYP2U1 sequences (237 *de novo* (GenBank: MZ383202–MZ383439) and 159 from public database) (Table S1) were done using ClustalX 2.1 (*Larkin et al., 2007*) in MEGA version 6.06 (*Tamura et al., 2013*).

## Genetic diversity

All the sequences were compared, and haplotypes identified using DnaSP 5.10.1(*Librado & Rozas, 2009*). Further, we calculated genetic diversity across all studied populations in terms of Number of haplotypes (nHT), Haplotype diversity (HTdiv), Nucleotide diversity (ndiv) together with their respective standard deviations using Arlequin version 3.5.2.2 (*Excoffier & Lischer, 2010*).

## Phylogenetic tree analyses and haplotype network

We investigated the association of Nigerian Muscovy duck with other Muscovy duck samples retrieved from public database (Table S1). MEGA version 6.06 (*Tamura et al., 2013*) was used to construct a rooted neighbor-joining (NJ) tree (*Saitou & Nei, 1987*) with the Maximum Composite Likelihood evolutionary distance approach (*Tamura, Nei & Kumar, 2004*). The bootstrap test was employed at 1,000 replications to assess the confidence of each node (*Felsenstein, 1985*). Further, we visualized the relationships between haplotypes by constructing median-joining networks of 397 Muscovy duck cytochrome b (rooted with chicken sequence; GenBank: KM886937) and 396 Muscovy duck nuclear DNA CYP2U1 sequences. Both networks were constructed using the median-joining (MJ) algorithm (*Bandelt, Forster & Röhl, 1999*) implemented in Network 5.0.1.1 (www.fluxus-engineering.com). The Median networks with all possible short trees were simplified by running the maximum parsimony (MP) calculation option to eliminate superfluous nodes and links.

**Table 1 Genetic diversity estimates of Nigerian Muscovy populations.**

| Population | Size[1] | Size[2] | nHT[1] | nHT[2] | HTdiv (SD)[1] | HTdiv (SD)[2] | ndiv (SD)[1] | ndiv (SD)[2] | D[1] | D[2] | Fs[1] | Fs[2] |
|---|---|---|---|---|---|---|---|---|---|---|---|---|
| BAYELSA | 62 | 67 | 9 | 28 | 0.373 (0.078) | 0.859 (0.038) | 0.054 (0.044) | 0.146 (0.079) | −2.206* | −0.980 | −4.758* | −10.355* |
| EKITI | 43 | 36 | 1 | 11 | – | 0.744 (0.066) | – | 0.047 (0.027) | – | 2.411 | – | −0.679 |
| KANO | 57 | 52 | 8 | 15 | 0.347 (0.080) | 0.714 (0.065) | 0.051 (0.046) | 0.040 (0.023) | −2.209* | −0.762 | −4.985* | −3.240 |
| OYO | 66 | 71 | 3 | 32 | 0.060 (0.040) | 0.869 (0.037) | 0.030 (0.074) | 0.053 (0.029) | −1.432* | −1.396* | −3.423* | −18.418* |
| SOKOTO | 15 | 8 | 3 | 6 | 0.257 (0.142) | 0.893 (0.111) | 0.133 (0.109) | 0.030 (0.021) | −2.040* | −1.320 | 0.780 | −2.015* |
| TARABA | 5 | 3 | 1 | 2 | – | 0.667 (0.314) | – | 0.032 (0.028) | – | 0.000 | – | 2.022 |
| NIGER[3] | 14 | 15 | 4 | 8 | 0.396 (0.159) | 0.791 (0.105) | 0.163 (0.132) | 0.054 (0.032) | −1.728* | −0.537 | −0.469 | −0.460 |
| KWARA[3] | 57 | 64 | 14 | 24 | 0.458 (0.083) | 0.775 (0.055) | 0.080 (0.048) | 0.039 (0.022) | −2.086* | −1.948* | −3.184 | −12.551* |
| POP EXP.[4] | 290 | 170 | 15 | 8 | 0.120 (0.026) | 0.358 (0.045) | 0.011 (0.014) | 0.107 (0.108) | −2.409* | −0.707 | −25.209* | −5.228* |
| NIGERIA | 319 | 316 | 31 | 91 | 0.266 (0.033) | 0.800 (0.023) | 0.016 (0.012) | 0.050 (0.028) | −2.605* | −2.022* | −29.584* | −25.288* |
| INDIA[5] | 78 | 80 | 11 | 9 | 0.706 (0.044) | 0.596 (0.057) | 0.292 (0.172) | 0.126 (0.078) | 1.170 | −1.095 | −0.233 | −0.185 |

**Note:**

Total number of samples in each region; nHT, Number of haplotypes; HTdiv (SD), Haplotype diversity (standard deviation); ndiv, Nucleotide diversity (standard deviation); $D$, Tajima's $D$ test of selective neutrality; $Fs$, Fu's $Fs$ test of selective neutrality; [1]Cytochrome b; [2]CYP2U1; [3]Retrieved from public database (*Sola-Ojo et al., 2021*); [4]Nigerian Muscovyduck population that showed the star-like pattern in Fig. 1B; [5]Retrieved from public database (*Kameshpandian, Thomas & Nagarajan, 2018*). Numbers with asterisks are statistically significant at 5% level.

## Demographic dynamic profiles and population genetic structure

The demographic patterns and population dynamics, demographic statistical parameters for Tajima's $D$ (*Tajima, 1989*), Fu's $Fs$ (*Fu, 1997*) tests, and population $F_{ST}$ of Nigerian Muscovy duck population were calculated using Arlequin version 3.5.2.2 (*Excoffier & Lischer, 2010*). The relationship between geographic distance and genetic distance was analyzed using linear regression model (LM) carried out under R statistical environment (*R Core Team, 2021*). Further, mismatch distribution patterns were estimated (*Rogers & Harpending, 1992*) with their geographical regions. We inferred genetic variation within populations, among populations, and groups of populations, analysis of molecular variance (AMOVA) with 50,000 permutations in Arlequin version 3.5.2.2 software. This analysis was done for Nigerian Muscovy duck populations at different hierarchical levels using the $F_{ST}$ parameter at a significant $P$ level of 0.05.

## RESULTS

### Cytochrome b genetic diversity

The 397 Muscovy duck cytochrome b sequences from Nigeria and India classify into 42 haplotypes (Table S2). The 319 Nigerian individuals (248 *de novo* (GenBank: MZ383528–MZ383776) and 71 from the GenBank) constitute 31 distinct haplotypes (Table S3), resulting from 57 polymorphic sites. The first most frequent haplotype, H4 (ADE15) occurs only in Nigerian samples while the second, H8 (BAA106) appears both in Nigerian and Indian populations (Table S2). The estimated haplotype diversity (HTdiv) across all Nigerian individuals is 0.266 ± 0.033, which is lower than the Indian estimate in almost three folds (Table 1). Within Nigeria, fourteen haplotypes occur in the Kwara population with the diversity value 0.458 ± 0.083, while three in the Oyo population with a diversity value of 0.060 ± 0.040 (Table 1). There were differences among populations with respect to their nucleotide compositions.

**Table 2 Pair wise FST values of Nigerian Muscovy populations.**

|  | BAYELSA | EKITI | KANO | OYO | SOKOTO | TARABA | KWARA | NIGER |
|---|---|---|---|---|---|---|---|---|
| BAYELSA | – | 0.012 | 0.011 | 0.005 | 0.008 | −0.077 | 0.058* | 0.031 |
| EKITI | 0.016* | – | 0.034 | 0.032* | 0.130* | 0.028 | 0.114* | 0.028 |
| KANO | −0.003 | 0.017 | – | 0.006 | 0.033 | −0.082 | 0.016 | 0.027 |
| OYO | 0.021* | −0.007 | 0.019* | – | 0.006 | −0.108 | 0.044* | 0.034 |
| SOKOTO | −0.012 | 0.079 | 0.003 | 0.088 | – | −0.090 | 0.044 | 0.125 |
| TARABA | −0.093 | 0.000 | −0.093 | −0.111 | −0.099 | – | −0.089 | −0.048 |
| KWARA | 0.039* | 0.035* | 0.029* | 0.050* | 0.006 | −0.079 | – | 0.034 |
| NIGER | 0.015 | 0.128* | 0.039* | 0.148* | −0.008 | −0.080 | 0.012 | – |

**Note:**
Above the diagonal $F_{ST}$ values of CYP2U1; below the diagonal $F_{ST}$ values of cytochrome b. *$F_{ST}$ P values significance level 0.05.

## Cytochrome b gene Phylogenetic tree and haplotype network

A Phylogenetic tree was constructed using 71 Muscovy duck sequences selected from haplotypes in Table S2. The 71 sequences are Nigerian Muscovy ($n = 42$) newly generated in this study together with other samples from Nigeria ($n = 16$), India ($n = 11$), the USA ($n = 1$), and China ($n = 1$) retrieved from the GenBank (Table S1). Chicken sequence (GenBank: KM886937) was used to root the phylogenetic tree. The phylogenetic tree showed significant clustering of Nigerian Muscovy duck together with Indian, USA, and Chinese Muscovy. A few minor clustering was also noticed within the major clade (Fig. S1).

The median-joining network of 397 Muscovy duck together with chicken cytochrome b revealed 42 different haplotypes, most of which were singleton (Fig. 1B (i)). Of the 42 haplotypes observed, H4 (ADE15) showed high frequency, which was present in all eight populations. The next prominent haplotype, H8 (BAA106) was present in five Nigerian Muscovy duck populations studied and shared values with Indian populations (Table S2). H4 (ADE15) exhibits a star-like pattern indicating rapid population expansion (Fig. 1B (i)).

## Cytochrome b population genetic structure and historical demographics

The Tajima's D (Tajima, 1989) and Fu's Fs (Fu, 1997) are −2.409 ($P < 0.05$) and −25.209 ($P < 0.05$), respectively (Table 1). These values also signal demographic expansion. The extent of haplotype-sharing in the network indicates the absence of a definite population structure in Nigeria Muscovy duck.

Mismatch distribution and pairwise $F_{ST}$ were calculated to infer the population dynamics and maternal genetic structure of Nigerian Muscovy duck. Mismatch distribution patterns were unimodal for Nigerian Muscovy duck populations (see Fig. S3), which is in accordance with the signal of demographic expansion. The $F_{ST}$ distance between most Nigerian Muscovy duck populations studied was low (Table 2). Among the eight populations, the highest genetic distance ($F_{ST}$) was observed between Oyo and Niger populations (Fig. 1A and Table 2), whereas the lowest (negative $F_{ST}$) were observed

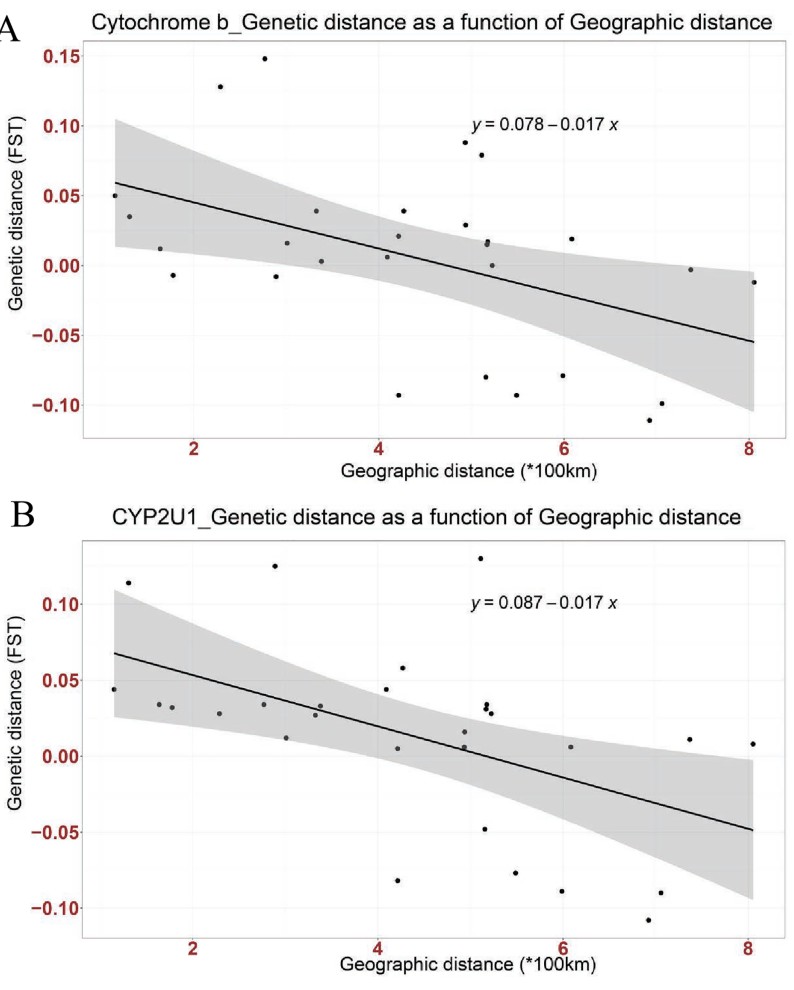

**Figure 2 Linear association between genetic distance ($F_{ST}$) and geographical distance ($^*$100 km) in two genes; (A) Cytochrome b, (B) CYP2U1.** The regression line shows negative correlation in both genes (Cytochrome b: Multiple R-squared: 0.2268, Adjusted R-squared: 0.197, *p*-value: 0.01042 at $P < 0.05$ and CYP2U1: Multiple R-squared: 0.2666, Adjusted R-squared: 0.2383, *p*-value: 0.004915 at $P < 0.005$).

between Taraba and the six populations except Ekiti, Niger and Sokoto, Bayelsa and Kano, Oyo and Ekiti. Negative $F_{ST}$ values is effectively seen as zero values. This indicates absence of genetic subdivision between the populations. The negative $F_{ST}$ may possibly be product of sampling size and low haplotype diversity. The regression analysis results showed a negative correlation between geographic distance and genetic distance ($R^2 = 0.2268$, $P < 0.05$), suggesting the possibility of "absence" or "insignificant change" in genetic distance associated with increase in geographic distance (Fig. 2A).

For the analysis of molecular variance (AMOVA) among populations, $F_{ST}$ was significant ($F_{ST} = 0.02550$, $P = 0.01075$; Table S6). Other analyses to assess differentiation within Nigeria were statistically non-significant ($P > 0.05$; Table S6).

## Nuclear DNA CYP2U1 gene genetic diversity

The nuclear DNA CYP2U1 gene sequences of 396 Nigerian and Indian Muscovy ducks classify into 95 haplotypes, and five haplotypes were common in both countries (Table S4). The 316 Nigerian individuals (237 *de novo* [GenBank: MZ383202–MZ383439] and 79 from the GenBank) constitute 91 distinct haplotypes (Table S5), resulting from 84 variable sites. The most frequent haplotype, H1 (ABO14) occurs only in Nigerian samples while the second, H82 (MUSCO02), occurs in Nigerian and Indian populations (Table S4). The estimated haplotype diversity (HTdiv) across all Nigerian individuals is 0.800 ± 0.023, which is higher than the Indian estimate in almost one and half folds (Table 1). Within Nigeria, 32 haplotypes occur in the Oyo population with a diversity value of 0.869 ± 0.037, while two in the Taraba population with a diversity value of 0.667 ± 0.314 (Table 1).

## Nuclear DNA CYP2U1 gene phylogenetic tree and haplotype network

The phylogenetic tree was constructed using 168 Muscovy duck nuclear DNA CYP2U1 sequences selected from haplotypes in Table S4. This comprises of Nigerian Muscovy duck sequences newly generated in this study together with other samples from India retrieved from the GenBank. The phylogenetic tree showed a major clustering of Nigerian Muscovy duck together with Indian Muscovy. A few minor clustering was also noticed within the major clade (Fig. S2).

The haplotype network of 396 Muscovy duck nuclear DNA CYP2U1 gene revealed 95 different haplotypes (Fig. 1B (ii)). Of the 95 haplotypes, H1 (ABO14) showed high frequency and was present in all eight populations (Table S4). The next prominent haplotype, H82 (MUSCO02), is dominated by Indian Muscovy duck populations and is also shared with Nigerian Muscovy duck from Oyo (Table S4). Some Nigerian populations showed the star-like pattern in Fig. 1B (ii), indicating rapid population expansion. The Tajima's *D* (*Tajima, 1989*) and Fu's *Fs* (*Fu, 1997*) are −0.707 ($P = 0.249$) and −5.228 ($P < 0.05$), respectively (Table 1). These values also signal demographic expansion. The extent of haplotype-sharing in the network indicates the absence of population structure in Nigeria.

## Nuclear DNA CYP2U1 gene population genetic structure and historical demographics

Mismatch distribution and pairwise $F_{ST}$ were calculated to infer the population dynamics and maternal genetic structure of Nigerian Muscovy duck. Mismatch distribution patterns were unimodal for Nigerian Muscovy duck populations (Fig. S3), which is in accordance with the signal of demographic expansion. The $F_{ST}$ distance between most Nigerian Muscovy duck sampling localities was low (Table 2). Among the eight populations, the highest genetic distance ($F_{ST}$) was observed between Ekiti and Sokoto populations (Table 2), whereas the lowest (negative $F_{ST}$) were observed between Taraba and the six populations except Ekiti. The negative $F_{ST}$ may possibly be product of sampling size and low haplotype diversity. The regression analysis results ($R^2 = 0.2666$, $P < 0.005$) also showed consistency with cytochrome b gene (Fig. 2B).

For the analysis of molecular variance (AMOVA) among populations, $F_{ST}$ was significant ($F_{ST}$ = 0.02318, $P$ = 0.00978; Table S7). Other analyses to assess differentiation within Nigeria were statistically non-significant ($P$ > 0.05; Table S7). Our findings with nuclear DNA CYP2U1 support the general absence of genetic structure in Nigeria as earlier revealed by the cytochrome b gene.

## DISCUSSION

In this study, the suitability of the nuclear DNA CYP2U1 gene was confirmed by assessment of variations in the genetic diversity and populations structure of Nigerian domestic Muscovy ducks in addition to the conventional mtDNA cytochrome b. High genetic diversity was revealed among Nigerian Muscovy duck sampling sites with 91 distinct haplotypes for the nuclear DNA CYP2U1 gene but low for cytochrome b with 31 haplotypes (Table 1; Tables S3 and S5). This discrepancy in the two markers possibly reflects sex bias reproduction among Nigerian Muscovy sampling sites. The low genetic diversity based on cytochrome b coincided with result of earlier studies on Muscovy duck populations in China and India (*Sun et al., 2012*; *Kameshpandian, Thomas & Nagarajan, 2018*). In a previous study on mitochondria D-loop, we also reported a low genetic diversity in Nigerian Muscovy duck populations (*Adeola et al., 2020*). The low genetic diversity in mtDNA cytochrome b of Nigerian Muscovy ducks corroborates previous reports that employed morphological parameters (*Oguntunji & Ayorinde, 2014*) and blood proteins (*Oguntunji & Ayorinde, 2015b*). An earlier attempt to classify three Nigerian Muscovy duck ecotypes using discriminant analysis of morpho-structural parameters reported clustering of the three ecotypes on canonical graph, low classification success rate (41.80–53.70%) and short Euclidean distance (2.010–3.758) among them (*Oguntunji & Ayorinde, 2014*). Similarly, the estimated genetic variability (heterozygosity) among them was similar (0.424–0.481); thus, indicating that the three ecotypes were under similar evolutionary forces and there were no genetic appreciable differences among them (*Oguntunji & Ayorinde, 2015b*). These studies adduced low genetic divergence among the investigated morphological ecotypes due to unrestricted gene flow between ducks of different ecotypes through inter-regional trade and human movement. Till date, agriculture remains the mainstay of the Nigerian economy and employs more than three-quarter of her workforce. It is noteworthy that preponderance of agricultural products is produced in northern Nigeria compared to the southern part and this elicited the age-long inter-regional trade involving daily transportation of huge number of unregistered and un-estimated number of both ruminant and non-ruminant animals to meet the social and nutritional needs of southern Nigerians. Hence, this study confirmed the low genetic diversity among Nigerian Muscovy duck populations (*Adeola et al., 2020*; *Sola-Ojo et al., 2021*). The phylogenetic tree showed significant clustering of Nigerian Muscovy duck and intermingled together with Indian, USA, and Chinese Muscovy ducks. A few minor clustering was also noticed within the major clade showing the genetic relationship of Nigerian Muscovy duck with others.

The median-joining network was constructed to visualize the relationships between haplotypes. In general, both markers revealed two groups in Nigerian Muscovy duck

populations. The first group (H4 (ADE15); cytochrome b and H1 (ABO14); CYP2U1) comprises individuals across all eight different locations in Nigeria. This suggested that one group of the Muscovy duck present in Nigeria descended from a single domestication. The most prominent haplotypes (H4 (ADE15); cytochrome b and H1 (ABO14); CYP2U1) with the highest frequencies exhibited a star-like pattern (Table 1; Fig. 1B). This pattern is often associated with a rapid demographic expansion probably from a small founder population (*Avise, 2000*). The signature of expansion is consistent with the significant negative values of Tajima's $D$ and Fu's $Fs$ tests ($P < 0.05$; Table 1) as well as the star-like pattern in the network of both markers (Fig. 1B). The unimodal mismatch distribution patterns further confirmed this in both markers (Fig. S3). Thus, the observed pattern in the first group of Nigerian Muscovy duck populations is likely due to a recent demographic expansion. The second group comprising Nigerian Muscovy ducks clustered with Indian Muscovy duck populations, suggesting that Muscovy duck present in Nigeria descended from a common origin and their Indian counterparts.

The extent of haplotype-sharing in the network indicates the absence of a definite population structure in Nigerian Muscovy duck. This was supported by the low $F_{ST}$ distance between most Nigerian Muscovy duck populations being studied. The regression analysis results showed a negative correlation between genetic and geographical distance in the two genes (Figs. 2A and 2B), with the slope suggesting the possibility of "absence" or "insignificant change" in genetic distance in relations to increase in geographic distance. However, this is unexpected for natural populations, where we would expect an increase in genetic distance with increased geographic distance. These results therefore suggest that geographic distance may not have played a major role in restricting gene flow among populations of Muscovy ducks in Nigeria, and this may be because of human activities aiding geneflow among populations. The reported inverse relationship between geographical and genetic distances in the present study is not unexpected. This unexpected anomaly might not be unconnected with the unrestricted gene flow among Nigerian Muscovy ducks because of inter-regional trade between northern and southern Nigeria. Majority of Nigerian Muscovy ducks are reared in northern Nigeria and serve as the population base of Muscovy ducks found in southern Nigeria. In view of this, the uncontrolled inter-regional trade encouraged exchange of germplasm between the ducks found in the two regions. This implies that Muscovy ducks found in northern and southern Nigeria have common origin, genetically similar with little or no genetic difference and the expected genetic diversity due to geographical separation have been overshadowed through inter-regional trade.

The AMOVA test among populations for cytochrome b and nuclear DNA CYP2U1 showed that $F_{ST}$ was significant ($F_{ST}$ = 0.02550 and 0.02318, $P$ = 0.01075 and 0.00978; Tables S6 and S7 respectively). Other analyses to assess differentiation within Nigeria were not significant ($P > 0.05$; Tables S6 and S7). Both markers supported the general absence of genetic structure in Nigerian Muscovy duck sampling sites. There was an absence of geographic signal and lack of genetic structure that likely suggests the extensive genetic intermixing among Muscovy duck within the country. This pattern observed is in accordance with earlier findings in Nigerian chickens (*Adebambo et al., 2010*) and

Guineafowl populations (*Adeola et al., 2015*) proposed to be likely due to intensive genetic intermixing between populations resulting from human migrations and trading.

## CONCLUSIONS

In this study, the genetic diversity and relationship within and between Muscovy duck sampling sites from eight different states of Nigeria and their counterparts from outside Africa were investigated using the mtDNA and nuclear DNA markers. Both markers (nuclear DNA CYP2U1 and mtDNA cytochrome b) revealed two groups (especially shown in the haplotype networks) in Nigerian Muscovy duck populations: the first group consisting of only Nigerian Muscovy duck sampling sites while the second group consists of the Nigerian and the India Muscovy duck populations. Neutrality test results indicated that all Nigerian populations experienced recent population expansion and/or genetic hitchhiking. This result suggested that the genetic structure could be explained by founder effect. The absence of geographic signal in Nigerian Muscovy duck sampling sites suggests extensive genetic intermixing consistent with previously studied poultry species in Nigeria. Further, the suitability of the nuclear DNA CYP2U1 gene in the analysis of genetic diversity was confirmed.

## ACKNOWLEDGEMENTS

The authors appreciate all volunteers who assisted in sampling and blood collection during the field work; F.E.S laboratory members, especially the undergraduate project student of 2019/2020 session of the Animal Production Department, University of Ilorin, and The Management of Fair and Firm farm Tanke Oke-Odo, Ilorin, Nigeria.

### Funding

This work was supported by the Sino-Africa Joint Research Center, the Chinese Academy of Sciences (SAJC202103), the Animal Branch of the Germplasm Bank of Wild Species, and the Chinese Academy of Sciences (the Large Research Infrastructure Funding). The Chinese Academy of Sciences President's International Fellowship Initiative provided support to Adeniyi C. Adeola (2021FYB0006). The funders had no role in study design, data collection and analysis, decision to publish, or preparation of the manuscript.

### Grant Disclosures

The following grant information was disclosed by the authors:
Sino-Africa Joint Research Center, the Chinese Academy of Sciences: SAJC202103.
Animal Branch of the Germplasm Bank of Wild Species.
Chinese Academy of Sciences (the Large Research Infrastructure Funding).
The Chinese Academy of Sciences President's International Fellowship: 2021FYB0006.

### Competing Interests

The authors declare that they have no competing interests.

## Author Contributions

- Adeniyi C. Adeola conceived and designed the experiments, performed the experiments, analyzed the data, prepared figures and/or tables, authored or reviewed drafts of the paper, and approved the final draft.
- Foluke E. Sola-Ojo conceived and designed the experiments, performed the experiments, analyzed the data, prepared figures and/or tables, authored or reviewed drafts of the paper, and approved the final draft.
- Yusuf A. Opeyemi conceived and designed the experiments, performed the experiments, analyzed the data, authored or reviewed drafts of the paper, and approved the final draft.
- Abel O. Oguntunji performed the experiments, authored or reviewed drafts of the paper, and approved the final draft.
- Lotanna Micah Nneji performed the experiments, authored or reviewed drafts of the paper, and approved the final draft.
- Muslim K. Ewuola performed the experiments, authored or reviewed drafts of the paper, and approved the final draft.
- Semiu F. Bello performed the experiments, analyzed the data, prepared figures and/or tables, authored or reviewed drafts of the paper, and approved the final draft.
- Wasiu A. Olaniyi analyzed the data, authored or reviewed drafts of the paper, and approved the final draft.
- Adeosun T. Adesoji performed the experiments, authored or reviewed drafts of the paper, and approved the final draft.
- Alex P. Karuno analyzed the data, authored or reviewed drafts of the paper, and approved the final draft.
- Oscar J. Sanke conceived and designed the experiments, authored or reviewed drafts of the paper, and approved the final draft.
- Ebiakpo Lucky Daniel performed the experiments, authored or reviewed drafts of the paper, and approved the final draft.

## Animal Ethics

The following information was supplied relating to ethical approvals (*i.e.*, approving body and any reference numbers):

Research Guidelines for the Institutional Review Board of Kunming Institute of Zoology, Chinese Academy of Sciences

## Field Study Permissions

The following information was supplied relating to field study approvals (*i.e.*, approving body and any reference numbers):

University Ethical Review Committee, University of Ilorin, Nigeria.

## Data Availability

The sequences are available in the Supplemental File and at GenBank: MZ383528 to MZ383776; MZ383202 to MZ383439.

## Supplemental Information

Supplemental information for this article can be found online at http://dx.doi.org/10.7717/peerj.13236#supplemental-information.

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
