# Peer review of "Genetic diversity and population structure of muscovy duck (Cairina moschata) from Nigeria"

_PeerJ, doi:10.7717/peerj.13236_

## Round 0.1 · original submission · Major Revisions

I agree with Reviewers' criticisms; major revisions are required.

Reviewer 1 ·

Basic reporting

The manuscript presents several formatting errors, a missing reference, and some omissions in the citation format, as well as typos and lack of standardization in the use of symbols.
The introduction requires more work. For example, the last paragraph uses very general and broad ideas at the beginning and ends up being very specific. A better introduction can be seen in Sola-Ojo et al. 2021
The map requires more descriptive geographic and political information.
Haplotypes networks and phylogenetic trees need more informative labels and colors for haplotypes and specimens.
The conclusion is misinterpreting the important result of the absence of population structure found in the data.
The Acknowledgements paragraph is unclear.

Experimental design

The research question in this manuscript has been established in previously published articles (Adeola et al., 2020; Sola-Ojo et al., 2021). The new data verified the previous findings. However, the in-depth analysis of how the gene flow (direction and degree) reduces the population structure of this species within the country is still lacking. Demographic models (e.g., isolation-with-migration models, IM) can be a good alternative.

Validity of the findings

No comment

Annotated reviews are not available for download in order to protect the identity of reviewers who chose to remain anonymous.

Reviewer 2 ·

Basic reporting

This is an interesting genetic study of the Muscovy duck in Nigeria that despite their economic importance little is known about the genetic characterization of their domestic populations in the country. This study could provide valuable breeding material for the poultry industry in Nigeria and other countries in the world.

My main criticisms are:
1. Introduction: This section needs to expand the information about what is already known about population genetic analysis this species in other regions and the potential explanation of such pattern. The introduction highlight the importance of the CYTB gene, but not of the CYP2U1 gene. What are the expectations based on previous findings? Also, what is the intention of comparing samples of Nigeria with samples from India?

2. Methods: The objective of this study state that authors intend to evaluate different Muscovy duck populations from Nigeria and India, however detailed information about such comparison is not clear. Locations from India are not even shown in this section. Based on the Table 2, I uderstand that the authors only cited results from a previous published study of Muscovy duck populations from India, but are not part of the analysis of this paper. The information here has to be better clarified.

3. Results: Figures needs to be improved in terms of design, resolution and size. The map lacks critical geographic information (latitude, longitude, scale, geographic reference, etc.). Color patterns and labels shown in the map and haplotype networks are not easy to understand.

3. Discussion: Authors provide hypothesis about unrestricted gene flow between populations, but no analysis regarding this aspect has been analyzed in this study. I would suggest to include isolation-with-migration (IM) models in the analysis, since they include parameters for population size, time of population separation, and gene exchange between populations. Such analysis can also provide more information for proper discussion of demographic aspects in this study.

Experimental design

The experimental design shown in the methods does not properly align with the objective stated in the introduction. A better organization of the information has to be done.

Validity of the findings

I have no comments on this section.

Additional comments

I have no comments on this section.

·

Basic reporting

The authors reported a paper on genetic diversity and population structure of muscovy duck from Nigeria.
The topic of the work is interesting and focused on the genetic characterization of local breeds/populations. However, there are some points of concern to be clarified and that make the paper not acceptable in the present form.
General comments about the introduction section:
-please try to furnish some more information about the current census of this breed in Nigeria. If number of heads is not available also the number of farms could be useful.
-lines 63-64: the authors wrote that the previous studies about genetic characterization are insufficient. Please try to summarize the main findings of the previous studies in this section (in few lines), just to present the state of art.
-line 65, the authors cited the "conservation of this breed": if I understood well is not endangered in Nigeria, is it? Please clarify this aspect.
No information in the introduction section are reported about the CYP2U1 gene: this is a major weakness. Please try to supply information that can justify the use of this marker.

Experimental design

I have a major comment about the aim of the work: why the authors included samples from India in their study? Is there any ethnological reason? Any evolutive reason? As the authors correctly reported the origin of this breed is from Central and South America. This aspect should be very well explained, otherwise the inclusion of samples from India is not justified.
Specific comments:
-line 94: please specify how many males and females.
-line 98: if the authors collected 2 samples per flock the final number of samples should be even...
-lines 99-100: it is quite strange to maintain the FTA cards in ethanol or stored at -80°C. Usually the FTA cards are stored at room temperature.... Just a comment.
-line 105: please specify the percentage of agarose used to prepare the gel. Moreover, please supply Nanodrop technical information (model and company).
-line 129: for what reason the number of sequences among the two markers is different?
-line 134: it could be better to specify how many sequences from India and how many from Nigeria.
-line 138: please supply a reference for MEGA software.
-line 143: please add a dot at the end of the sentence.

Validity of the findings

Results section
Please be consistent in using acronyms and abbreviations, i.e.:
-lines 183 and 184: Hap_4 and Hap_8 but in lines 202 and 203 HT4 and HT8
-Fst is written correctly in line 217. Please revise this acronym in all the manuscript long (including tables).
-line 193: please clarify why the authors used the chicken sequence to root the pylogenetic tree.
-lines 240-243: this part could be moved to materials and methods section.
-line 329: be careful in speculate bottleneck effect: no bottleneck analysis was carried out in this study.

---

## Round 0.2 · Minor Revisions

I agree with Reviewers #1 and #2: the manuscript still needs improvements and corrections.

Reviewer 1 ·

Basic reporting

There are still several inconsistencies or mistakes in data and table formatting.

Figure 1
The information of the number of samples by locality from the previous first draft was good. I suggest bringing that back.
Even though Niger and Kwara are published data it is good to have those localities on the map.
An inset map, an overview of the main map, including Nigeria and India could be a good option to provide a better overview of the main map.

Table 1, Table S1 and S2
The number of samples (Size1) for table 1, S1, and S2 of Ekiti and Kwara is different for the haplotype ADE15

Table S1 and S2 CYTB haplotype
There is inconsistency in the formatting of the tables. The column headings use capitalize and non capitalized names. Also, the "Frequency" label is different. Some values are centered and in bold. The same looks like are happening with Table S3 and S4.

Experimental design

No comment

Validity of the findings

Expand the idea of ​​"human activities" in the discussion and conclusion.
This probably is the most important process in the lack of genetic structure in this domesticated species.

Annotated reviews are not available for download in order to protect the identity of reviewers who chose to remain anonymous.

Reviewer 2 ·

Basic reporting

The article has been significantly improved, although I have a few observations. Attached my general/specific comments:

1. Introduction: Even though authors highlighted the importance of the CYTB gene and CYP2U1 gene, authors mentioned an additional marker (mithocondrial D-loop) that is not even explored in this article.

Lines 91-92: Why is it relevant to mention the areas of expression of the CYP2U gene at the organs level in the article? If it is worth to mention, you should do the same for the other markers.

2. Discussion: There seems to be a misinterpretation of the regression analysis results when authors mention that there is a "significative negative relationship" between genetic and geographical distance on the two genes. Make sure you consider the slope value and the range of genetic distance into account in your discussion of this aspect.

3. Figures: Map in Fig 1 still has to be improved. Please have an inset of Africa to show where the sampling sites are at a large scale. Also, make sure you include critical geographic information (latitude, longitude, spatial reference system). Change the symbol shape of locations shown in the legends for Fig S1 and S2; use a circle shape instead of a pentagon.

Experimental design

I have no comments on this section.

Validity of the findings

I have no comments on this section.

Additional comments

I have no comments on this section.

·

Basic reporting

First of all I want to thanks the authors for their efforts in preparing this revised version that in my opinion is more understandable for the readers.

Experimental design

It is now well explained and complete.

Validity of the findings

The authors modified the manuscript according the reviewer's comments and the new version is improved and acceptable in the present form.

---

## Round 0.3 · accepted · Accept

Please satisfy the observations made by Rev#1 marked in the annotated manuscript.

Reviewer 1 ·

Basic reporting

Very good job improving the manuscript. I don't have any major concerns with it.
A few simple observations are marked in the annotated manuscript.

Experimental design

No comment

Validity of the findings

No comment

Annotated reviews are not available for download in order to protect the identity of reviewers who chose to remain anonymous.

Reviewer 2 ·

Basic reporting

I have no comments. I would like to acknowledge the authors for their efforts improving the manuscript.

Experimental design

I have no comments. I would like to acknowledge the authors for their efforts improving the manuscript.

Validity of the findings

I have no comments. I would like to acknowledge the authors for their efforts improving the manuscript.

Additional comments

I have no comments. I would like to acknowledge the authors for their efforts improving the manuscript.